# Winning the Lottery with Continuous Sparsification

**Pedro Savarese**[*]
TTI-Chicago
savarese@ttic.edu

**Hugo Silva**[*]
University of Alberta
hugoluis@ualberta.ca

**Michael Maire**
University of Chicago
mmaire@uchicago.edu

## Abstract

The search for efficient, sparse deep neural network models is most prominently performed by pruning: training a dense, overparameterized network and removing parameters, usually via following a manually-crafted heuristic. Additionally, the recent Lottery Ticket Hypothesis conjectures that, for a typically-sized neural network, it is possible to find small sub-networks which, when trained from scratch on a comparable budget, match the performance of the original dense counterpart. We revisit fundamental aspects of pruning algorithms, pointing out missing ingredients in previous approaches, and develop a method, Continuous Sparsification, which searches for sparse networks based on a novel approximation of an intractable $\ell_0$ regularization. We compare against dominant heuristic-based methods on pruning as well as ticket search – finding sparse subnetworks that can be successfully re-trained from an early iterate. Empirical results show that we surpass the state-of-the-art for both objectives, across models and datasets, including VGG trained on CIFAR-10 and ResNet-50 trained on ImageNet. In addition to setting a new standard for pruning, Continuous Sparsification also offers fast parallel ticket search, opening doors to new applications of the Lottery Ticket Hypothesis.

## 1 Introduction

Although deep neural networks have become ubiquitous in fields such as computer vision and natural language processing, extreme overparameterization is typically required to achieve state-of-the-art results, incurring higher training costs and hindering applications limited by memory or inference time. Recent theoretical work suggest that overparameterization plays a key role in network training dynamics [1] and generalization [2]. However, it remains unclear whether, in practice, overparameterization is truly necessary to train networks to state-of-the-art performance.

Concurrently, empirical approaches have been successful in finding compact neural networks, either by shrinking trained models [3, 4, 5] or through efficient architectures, yielding less overparameterized models that can be trained from scratch [6]. Recently, combining these two strategies has lead to new methods which discover efficient architectures through optimization instead of design [7, 8]. Nonetheless, parameter efficiency is typically maximized by pruning an already trained network.

Despite the fact that the search for sparse solutions to optimization problems can be naturally described by $\ell_0$ regularization, the vast majority of pruning methods rely on manually-designed strategies that are not based on the $\ell_0$ penalty [3, 4, 9, 10]. The approaches that aim to approximate an $\ell_0$-regularized problem in order to find sparse, less overparameterized networks are limited in number [11, 12] and fail to perform competitively against heuristic-based pruning methods.

Prior work has shown that pruned networks are hard to train from scratch [3], suggesting that while overparameterization is not necessary for a model's capacity, it might be required for successful training. Frankle and Carbin [13] put this idea into question by training heavily pruned networks

---

[*]equal contribution

from scratch, while achieving performance matching that of their original counterparts. A key finding is that the same initialization should be used when re-training the pruned network, or, equivalently, that better strategies – depending on *future weights* – can result in trainable pruned networks.

More recently, Frankle *et al.* [14] show that although this approach can fail in large-scale settings, pruned networks can be successfully re-trained when parameters from very early training are used as initialization. Coupling a pruned network with a set of parameter values from initialization yields a *ticket* – a *winning ticket* if it is able to match the dense model's performance when trained in isolation for a comparable number of iterations. These have already found applications in, for example, transfer learning [15, 16, 17], making *ticket search* a problem of independent interest.

Iterative Magnitude Pruning (IMP) [13], the first and currently only algorithm able to find winning tickets, consists of a repeating a two-stage procedure that alternates between training and pruning. IMP relies on a sensible choice for pruning strategy [18] and can be costly: maximizing the performance of the found subnetworks typically requires multiple *rounds* of training followed by pruning [19].

In this paper, we focus on two questions related to pruning and the Lottery Ticket Hypothesis. First, can we find sparse networks with competitive performance by approximating $\ell_0$ regularization instead of relying on a heuristic pruning strategy and, if yes, what are the missing ingredients in previous approaches [11, 12]? Second, would a method that relies on $\ell_0$-regularization, rather than an ad-hoc heuristic, be able to find winning tickets, as IMP does?

We provide positive answers to both questions by proposing Continuous Sparsification[1], a new pruning method that relies on approximating the intractable $\ell_0$ penalty and finds networks that perform competitively when either fine-tuned or re-trained. Unlike prior $\ell_0$-based approaches, our approximation is *deterministic*, providing insights and raising questions on how pruning and sparse regularization should be performed. The core of our method lies in constructing a smooth continuation path [20] connecting training of soft-gated parameters and the intractable $\ell_0$-regularized objective.

**Contributions:**

- We propose a novel approximation to $\ell_0$ regularization, resulting in Continuous Sparsification, a new pruning method with theoretical and empirical advantages over previous $\ell_0$-based approaches. We show through experiments that the deterministic nature of our re-parameterization is key to achieving competitive results with $\ell_0$ approximations.

- We show that Continuous Sparsification outperforms state-of-the-art heuristic-based pruning methods. Our experiments include pruning of VGG-16 [21] and ResNet-20 [22] trained on CIFAR-10 [23], and ResNet-50 trained on ImageNet [24].

- Our method raises questions on how to do better ticket search – producing subnetworks that can be re-trained from early iterates. We show empirically that Continuous Sparsification is capable of finding subnetworks of VGG-16, ResNet-20, and ResNet-50 that, when re-trained, outperform ones found by IMP. Moreover, the search cost of our method does not depend on the produced subnetwork's sparsity, making ticket search considerably more efficient when run in parallel.

## 2 Preliminaries

Here we define terms used throughout the paper.

**Subnetwork:** For a network $f$ that maps samples $x \in \mathcal{X}$ and parameters $w \in \mathbb{R}^d$ to $f(x; w)$, a subnetwork $f'$ of $f$ is given by a binary mask $m \in \{0, 1\}^d$, where a parameter component $w_i$ is kept in $f'$ if $m_i = 1$ and removed otherwise *i.e.*, $f' : x, w \mapsto f(x; w \odot m)$, with $\odot$ denoting element-wise multiplication. For any configuration $m$, the effective parameter space of the induced network $f'$ is $\{w \odot m | w \in \mathbb{R}^d\}$ – a $\|m\|_0$-dimensional space, hence we say that the subnetwork $f'$ has $\|m\|_0$ many parameters instead of $d$.

**Matching subnetwork:** For a network $f$ and randomly-initialized parameters $w^{(0)}$, a matching subnetwork $f'$ of $f$ is given by a configuration $m \in \{0, 1\}^d$, such that $f'$ can be trained in isolation from $w'^{(0)} = w^{(k)} \odot m$, where $w^{(k)}$ is the collection of parameter values obtained by training $f$ from $w^{(0)}$ for $k$ iterations, where $k$ is small. Moreover, to be a matching subnetwork, $f'$ needs to match the performance of a trained $f$ given the same budget, when measured in terms of training iterations.

**Winning ticket:** For a network $f$ and randomly-initialized parameters $w^{(0)}$, a winning ticket is a matching subnetwork $f'$ of $f$ that can be trained in isolation *from initialization, i.e.,* $w'^{(0)} = w^{(0)} \odot m$. In other words, a winning ticket is a matching subnetwork such that $k = 0$ in the definition above.

**Ticket search** is the task of finding matching subnetworks given a network $f$ and randomly-initialized parameters $w^{(0)}$. We say that an algorithm $A$ performs ticket search if $A(f, w^{(0)}) = m \in \{0,1\}^d$, such that $m$ induces a (possibly matching) subnetwork $f'$.

## 3 Related Work

### 3.1 Sparse Networks

Classical pruning methods [25] follow a pre-defined strategy to remove weights, and generally operate by ranking parameters according to an easy-to-compute statistic like weight magnitude [3]. Such methods rely on the assumption that the considered statistic is a sensible surrogate for how much each parameter affects a network's output, and typically select weights for removal once the dense model has been fully trained. Magnitude-based pruning, the most prominent heuristic pruning method, improves when given multiple *rounds* of training followed by pruning [3, 26].

Another approach consists of approximating an intractable $\ell_0$-regularized objective which accounts for the number of non-zero weights in the model, yielding one-stage procedures that can be fully described in the optimization framework. More common in the literature are stochastic approximations, where a binary mask $m$ over the weights is sampled from a distribution $\mathcal{M}(s)$ at each training iteration, introducing new variables $s$ which are optimized jointly with the weight parameters [11, 12].

Training the mask parameters $s$ is done by estimating the gradients of the *expected* loss w.r.t. $s$, *e.g.,* via the straight-through estimator [27], thus relying on estimated gradients which can be biased and have high variance. $\ell_0$-based methods have the advantage of not relying on a heuristic to prune weights, and continuously sparsify the network *during* training instead of at pre-defined steps.

### 3.2 Lottery Ticket Hypothesis

Frankle and Carbin [13] show that, in some settings, sparse subnetworks can be successfully re-trained and yield better performance than their original dense networks, often also on a smaller compute budget for re-training. This observation leads to the Lottery Ticket Hypothesis [13], which conjectures that for a reasonably-sized network $f$ and randomly-initialized parameters $w^{(0)} \in \mathbb{R}^d$, there exists a sparse subnetwork $f'$, given by a configuration $m \in \{0,1\}^d$, $\|m\|_0 \ll d$, that can be trained from $w'^{(0)} = m \odot w^{(0)}$ to perform comparably to a trained version of the original model $f$.

The proposal of Iterative Magnitude Pruning (IMP; Algorithm 1) [19] supports this hypothesis. IMP is capable of finding such subnetworks, named winning tickets, in convolutional networks trained for image classification. IMP operates in multiple *rounds*, sparsifying the network at discrete time intervals and producing subnetworks with increasing sparsity levels during execution. More specifically, each *round* in IMP consists of: (1) training the weights $w$ of a network, (2) pruning a fixed fraction of the weights with the smallest magnitude, and (3) *rewinding*: setting the remaining weights back to their original initialization $w^{(0)}$.

Following Frankle *et al.* [19], we consider a general form of IMP where step (3) is relaxed to rewind the weights to an early iterate $w^{(k)}$ (for relatively small $k$) instead of the original initialization values $w^{(0)}$. We refer to the process of searching for a sparse subnetwork and a set of early iterates $w^{(k)}$ as ticket search, even though the produced subnetworks are truly only winning tickets when they perform comparably to the dense model when trained in isolation from $w^{(0)}$, *i.e.,* $k = 0$.

The search for winning tickets has attracted attention due to their valuable properties. In small-scale settings, tickets can be trained *faster* than their dense counterparts while yielding *better* final performance [13]. Moreover, they can be transferred between datasets [16, 17] and training methods [15]. Zhou *et al.* [18] attempt to better understand the Lottery Ticket Hypothesis through extensive experiments, showing that a stochastic approximation to $\ell_0$ regularization can be used to perform ticket search with SGD, without ever training the weights (non-retroactive search).

| **Algorithm 1** Iterative Magnitude Pruning [19] | **Algorithm 2** Continuous Sparsification |
|---|---|
| **Input:** Pruning ratio $\tau$, number of rounds $R$, iterations per round $T$, rewind point $k$ | **Input:** Mask init $s^{(0)}$, penalty $\lambda$, number of rounds $R$, iterations per round $T$, rewind point $k$ |
| 1: Initialize $w \sim \mathcal{D}$, $m \leftarrow \vec{1}^d$, $r \leftarrow 1$ | 1: Initialize $w \sim \mathcal{D}$, $s \leftarrow s^{(0)}$, $\beta \leftarrow 1$, $r \leftarrow 1$ |
| 2: Minimize $L(f(\,\cdot\,; m \odot w))$ for $T$ iterations, producing $w^{(T)}$ | 2: Minimize $L(f(\,\cdot\,; \sigma(\beta s) \odot w)) + \lambda \,\|\sigma(\beta s)\|_1$ for $T$ iterations while increasing $\beta$, producing $w^{(T)}$, $s^{(T)}$, and $\beta^{(T)}$ |
| 3: Remove $\tau$ percent of the weights with smallest magnitude | 3: If $r = R$, output $f(\,\cdot\,; H(s^{(T)}) \odot w^{(k)})$ |
| 4: If $r = R$, output $f(\,\cdot\,; m \odot w^{(k)})$ | 4: Otherwise, set $s \leftarrow \min(\beta^{(T)} s^{(T)}, s^{(0)})$, $\beta \leftarrow 1$, $r \leftarrow r + 1$ and go back to step 2, thereby starting a new round |
| 5: Otherwise, set $w \leftarrow w^{(k)}$, $r \leftarrow r + 1$ and go back to step 2, thereby starting a new round | |

# 4 Method

Our goal is to design a method that can efficiently sparsify networks without causing performance degradation. Ideally, and in contrast to magnitude pruning, the time to produce a subnetwork should be independent of its sparsity. Unlike dominant pruning approaches [3, 4, 9], we rely on approximating $\ell_0$ regularization, as it induces a clear trade-off between sparsity and performance, providing a way to maximize sparsity while maintaining performance. By continuously sparsifying the network during training, we do not require a heuristic to select which parameters to remove or when to remove them.

To avoid gradient estimators and to avoid having to *commit* to a configuration for $m$ to be used at inference – obstacles that are inherent to stochastic approximations to the $\ell_0$ objective – we design a *deterministic* approximation instead, as we describe below.

## 4.1 Continuous Sparsification by Learning Deterministic Masks

Given a network $f$ that maps samples $x$ to $f(x; w)$ using parameters $w \in \mathbb{R}^d$, we first frame the search for sparse subnetworks as a loss minimization problem with $\ell_0$ regularization:

$$\min_{w \in \mathbb{R}^d} \quad L(f(\,\cdot\,; w)) + \lambda \cdot \|w\|_0 \,, \tag{1}$$

where $L(f(\,\cdot\,; w))$ denotes the loss incurred by the network $f(\,\cdot\,; w)$ and $\lambda \geq 0$ controls the trade-off between the loss and number of parameters $\|w\|_0$. We restate the above minimization problem as

$$\min_{w \in \mathbb{R}^d, \, m \in \{0,1\}^d} \quad L(f(\,\cdot\,; m \odot w)) + \lambda \cdot \|m\|_1 \,, \tag{2}$$

which uses the fact that $\|m\|_0 = \|m\|_1$ for binary $m$. While the $\ell_1$ penalty is amenable to subgradient descent, the combinatorial constraint $m \in \{0,1\}^d$ makes local search unsuited for the problem above.

As in most methods that approximate $\ell_0$ regularization, we will circumvent the discrete space of $m$ by re-parameterizing it as a function of a newly-introduced variable $s \in \mathbb{R}^d$. In contrast to previous work [12, 11], we propose a re-parameterization that is fully *deterministic*, hence avoiding biased and/or noisy training caused by gradient estimators [27].

Consider an intermediate and still intractable problem, given by defining $m := H(s)$, with $s \in \mathbb{R}^d_{\neq 0}$ and $H : \mathbb{R}_{\neq 0} \to \{0, 1\}$ being the Heaviside step function applied element-wise, *i.e.,* $H(s) = 1$ if $s > 0$ and $0$ otherwise. This yields the following equivalent form for the problem in (2):

$$\min_{w \in \mathbb{R}^d, \, s \in \mathbb{R}^d_{\neq 0}} \quad L(f(\,\cdot\,; H(s) \odot w)) + \lambda \cdot \|H(s)\|_1 \,. \tag{3}$$

Being equivalent to (1), the above is still intractable: the step function $H$ is discontinuous and its derivative is zero everywhere. We approximate $H$ by constructing a set of functions indexed by $\beta \in [1, \infty)$ given by $s \mapsto \sigma(\beta s)$ where $\sigma$ is the sigmoid function $\sigma(s) = \frac{1}{1+e^{-s}}$ applied element-wise. This set can be seen as a *path* parameterized by $\beta$, and given any fixed $s \in \mathbb{R}_{\neq 0}$, we have at one of

its endpoints $\lim_{\beta \to \infty} \sigma(\beta s) = H(s)$. Conversely, for $\beta = 1$ we have $\sigma(\beta s) = \sigma(s)$, the standard sigmoid activation function that is smooth and widely used in neural network models.

Using this family of functions to approximate $H$ yields the re-parameterization $m := \sigma(\beta s)$. Controlling the inverse temperature $\beta$ allows interpolation between the sigmoid activation $\sigma(s)$, which assigns continuous values for $m$, and the step function $H(s) \in \{0, 1\}$. Each $\beta$ induces the objective

$$L_\beta(w, s) := L(f(\,\cdot\,; \sigma(\beta s) \odot w)) + \lambda \cdot \|\sigma(\beta s)\|_1 . \tag{4}$$

Note that if $L$ is continuous in $w$, then:

$$\min_{\substack{w \in \mathbb{R}^d \\ s \in \mathbb{R}^d_{\neq 0}}} \lim_{\beta \to \infty} L_\beta(w, s) = \min_{\substack{w \in \mathbb{R}^d \\ s \in \mathbb{R}^d_{\neq 0}}} L(f(\,\cdot\,; H(s) \odot w)) + \lambda \cdot \|H(s)\|_1 , \tag{5}$$

where the right-hand-side is equivalent to the $\ell_0$-regularized objective. Therefore, $\beta$ controls the computational hardness of the objective: as $\beta$ increases from 1 to $\infty$, the objective changes from $L_1(w, s)$, where a soft-gating $w \odot \sigma(s)$ is applied to the weights, to $L_\infty(w, s)$, where weights are either removed or fully preserved. Increasing the hardness of the underlying objective during training stems from continuation methods [20] and can be successful in approximating intractable problems.

In terms of sparsification, every negative component of $s$ will drive the corresponding component of $w \odot \sigma(\beta s)$ to 0 as $\beta \to \infty$, effectively pruning a weight. While analytically it is never the case that $\sigma(\beta s) = 0$ regardless of how large $\beta$ is, limited numerical precision has a fortunate side-effect of causing actual sparsification to the network during training as $\beta$ becomes sufficiently large.

In a nutshell, our method consists of learning sparse networks by minimizing $L_\beta(w, s)$ for $T$ parameter updates with gradient descent while jointly annealing $\beta$: producing $w^{(T)}$, $s^{(T)}$ and $\beta^{(T)}$. Note that, in order to recover a binary mask $m$ from our re-parameterization, $\beta$ must be large enough such that, *numerically*[2], $\sigma(\beta^{(T)} s^{(T)}) = H(s^{(T)})$. Alternatively, we can directly output $m = H(s^{(T)})$ at the end of training, guaranteeing that the learned mask is indeed binary. We adopt an exponential schedule $\beta^{(t)} = (\beta^{(T)})^{\frac{t}{T}}$ for $\beta$ during training, increasing it from 1 up to $\beta^{(T)}$. Such a schedule has the advantage of only requiring us to tune $\beta^{(T)}$, and has been successfully utilized in prior work [28].

## 4.2 Ticket Search through Continuous Sparsification

The method described above is essentially a pruning method, which we use to replace magnitude-based pruning as the backbone for ticket search. Note that searching for matching subnetworks requires produced masks to be binary: otherwise, the magnitude of the weights will also be learned. We guarantee that the final mask is binary regardless of numerical precision by outputting $H(s^{(T)})$.

Similarly to IMP, our ticket search procedure operates in *rounds*, where each round consists of training and sparsifying the network. At the beginning of a round, we set $\beta$ back to 1 so that additional weights can be removed (otherwise $\beta$ would be large throughout the round, causing a vanishing Jacobian of $\sigma(\beta s)$ w.r.t. $s$). Moreover, we reset the parameter $s$ of each weight $w$ that has not been suppressed by the optimizer during the round (*i.e.,* weights whose gating value has increased during training). This is achieved by setting $s \leftarrow \min(\beta^{(T)} s^{(T)}, s^{(0)})$, effectively resetting the soft mask parameters $s$ for "kept" weights without interfering with weights that have been suppressed. Algorithm 2 presents our method for ticket search, which does **not** rewind weights between rounds, in contrast to IMP [19].

## 4.3 Comparison to Stochastic Approaches

Prior works [11, 12] approximating the $\ell_0$ objective adopt a stochastic re-parameterization $m \sim \mathcal{M}(s)$ for some distribution $\mathcal{M}$ with parameters $s$. During training, a new binary mask $m$ is sampled from $\mathcal{M}(s)$ at every forward pass of the network. Hence, outputs can change drastically from one pass to another due to variance in sampling. Such approaches have found limited success in pruning.

Gale *et al.* [26] report that the stochastic approach from Louizos *et al.* [12] fails to sparsify a residual network without degrading its accuracy to random chance. Stochastic approximations introduce

another problem: different behavior between training and inference. While a new mask is sampled at each training iteration, at inference it is common to use a deterministic mask, such as that with highest mass [11] or an approximation for it [12]. This assures that the outputs at inference are consistent, but can introduce a gap in sparsity and performance between training and inference modes.

Conversely, Continuous Sparsification offers consistency in training mode – outputs for a input are the same across forward passes – and no gap between training and inference. In Section 5.2, experimental comparisons with stochastic approximations show that these differences play a key role in attaining faster training, higher sparsity, and superior performance when pruning deep networks.

## 5 Experiments

We compare methods on the tasks of pruning and finding matching subnetworks. We quantify the performance of ticket search by focusing on two specific subnetworks produced by each method:

- **Sparsest matching subnetwork**: the sparsest subnetwork that, when trained in isolation from an early iterate, yields performance no worse than that achieved by the trained dense counterpart.
- **Best performing subnetwork**: the subnetwork that achieves the best performance when trained in isolation from an early iterate, regardless of its sparsity.

We also measure the efficiency of each method in terms of total number of epochs to produce subnetworks, *given enough parallel computing resources*. As we will see, Continuous Sparsification is particularly suited for parallel execution since it requires relatively few rounds to produce subnetworks regardless of sparsity. On the other hand, CS offers no explicit mechanism to control the sparsity of the found subnetworks, hence producing a subnetwork with a pre-defined sparsity level can require multiple runs with different hyperparameter settings. For this use case, IMP is more efficient by design, since a single run suffices to produce subnetworks with varying, pre-defined sparsity levels.

For Continuous Sparsification, we set hyperparameters $\lambda = 10^{-8}$ and $\beta^{(T)} = 200$, based on analysis in Appendix A, which studies how $\lambda$, $s^{(0)}$, and $\beta^{(T)}$ affect the sparsity of produced subnetworks. We observe that $s^{(0)}$ has a major impact on sparsity levels, while $\lambda$ and $\beta^{(T)}$ require little to no tuning.

We reiterate that Continuous Sparsification does not perform weight rewinding in the following experiments; rather, it maintains weights between rounds. Our experimental comparisons include a variant of IMP that also does not rewind weights between rounds, which we denote as "continued" IMP (IMP-C). Algorithms 1 and 2 provide more implementation details. Comparisons against a baseline inspired by Zhou *et al.* [18], and described in Appendix B, on the tasks of learning a supermask and ticket search on a 6-layer CNN can be found in Appendices C and D.

### 5.1 Ticket Search on Residual Networks and VGG

First, we evaluate how IMP and CS perform on the task of ticket search for VGG-16 [21] and ResNet-20[3] [22] trained on the CIFAR-10 dataset, a setting where IMP can take over 10 rounds (850 epochs given 85 epochs per round [19]) to find sparse subnetworks. We follow Frankle and Carbin's setup [13]: in each round, we train with SGD, a learning rate of 0.1, and a momentum of 0.9, for a total of 85 epochs, using a batch size of 64 for VGG and 128 for ResNet. We decay the learning rate by a factor of 10 at epochs 56 and 71, and utilize a weight decay of 0.0001.

For CS, we do not apply weight decay to the mask parameters $s$, since they are already suffer $\ell_1$ regularization. Sparsification is performed on all convolutional layers, excluding the two skip-connections of ResNet-20 that have $1 \times 1$ kernels: for IMP, their parameters are not pruned, while for CS their weights do not have an associated learnable mask.

We evaluate produced subnetworks by initializing their weights with the iterates from the end of epoch 2, similarly to Frankle *et al.* [19], followed by re-training. IMP performs global pruning at a per-round rate of removing 20% of the remaining parameters with smallest magnitude. We run IMP for 30 iterations, yielding 30 tickets with varying sparsity levels (80%, 64%, . . . ). To produce tickets of differing sparsity with CS, we vary $s^{(0)}$ across 11 values from $-0.3$ to $0.3$, performing a run of 5 rounds for each setting. We repeat experiments 3 times, with different random seeds.

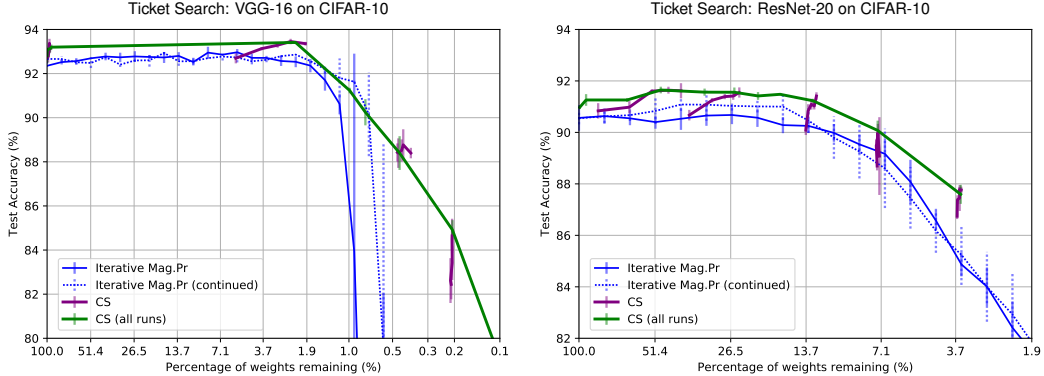

Figure 1: Test accuracy and sparsity of subnetworks produced by IMP and CS after re-training from weights of epoch 2. Purple curves show individual runs of CS, while the green curve connects tickets produced after 5 rounds of CS with varying $s^{(0)}$. Iterative Magnitude Pruning (continued) refers to IMP without rewinding between rounds. Error bars depict variance across 3 runs.

Table 1: Test accuracy and sparsity of the sparsest matching and best performing subnetworks produced by CS, IMP, and IMP-C (IMP without rewinding) for VGG-16 and ResNet-20 trained on CIFAR-10.

|  |  | VGG-16 | | | ResNet-20 | | |
| Method | | Round | Test Accuracy | Weights Remaining | Round | Test Accuracy | Weights Remaining |
| Dense Network | | 1 | 92.35% | 100.0% | 1 | 90.55% | 100.0% |
| Sparsest | IMP | 18 | 92.36% | 1.8% | 7 | 90.57% | 20.9% |
| Matching | IMP-C | 18 | 92.56% | 1.8% | 8 | 91.00% | 16.7% |
| Subnetwork | CS | 5 | 93.35% | **1.7%** | 5 | 91.43% | **12.3%** |
| Best | IMP | 13 | 92.97% | 5.5% | 6 | 90.67% | 26.2% |
| Performing | IMP-C | 12 | 92.77% | 6.9% | 4 | 91.08% | 40.9% |
| Subnetwork | CS | 4 | **93.45%** | 2.4% | 5 | **91.54%** | 16.9% |

Figure 1 shows the performance and sparsity of tickets produced by CS and IMP, including IMP without rewinding (continued). Purple curves show individual runs of CS for different values of $s^{(0)}$, each consisting of 5 rounds, and the green curve shows the performance of subnetworks produced with different hyperparameters. Plots of individual runs are available in Appendix E, but have been omitted here for the sake of clarity. Given a search budget of 5 rounds (*i.e.,* $5 \times 85 = 425$ epochs), CS successfully finds subnetworks with diverse sparsity levels. Notably, IMP produces tickets with superior performance when weight rewinding is not employed between rounds.

Table 1 summarizes the performance of each method when evaluated in terms of the sparsest matching and best performing subnetworks. IMP-C denotes IMP without rewinding, *i.e.,* IMP (continued) from Figure 1. Sparsest matching subnetworks produced by CS are sparser than the ones found by IMP and IMP-C, while also delivering higher accuracy. CS also outperforms IMP and IMP-C when evaluating the best performing produced subnetworks. In particular, CS yields highly sparse subnetworks that outperform the original model by approximately $1\%$ on both VGG-16 and ResNet-20.

If all runs are executed in parallel, producing all tickets presented in Figure 1 takes CS a total of $5 \times 85 = 425$ training epochs, while IMP requires $30 \times 85 = 2550$ epochs instead. Note that our re-parameterization results in approximately $10\%$ longer training times on a GPU due to the mask parameters $s$, therefore wall-clock time for CS is $10\%$ higher per epoch. Sequential search takes $5 \times 11 \times 85 = 4675$ epochs for CS to produce all tickets in Figure 1, while IMP requires $30 \times 85 = 2550$ epochs, hence CS is faster given sufficient parallelism, but slower if run sequentially. Appendix F shows preliminary results of a variant of CS designed for sequential search.

## 5.2 Pruning

Since CS is a general-purpose method to find sparse networks, we also evaluate it on the more standard task of network pruning, where produced subnetworks are fine-tuned instead of re-trained. We compare it against the prominent pruning methods AMC [29], magnitude pruning (MP) [3],

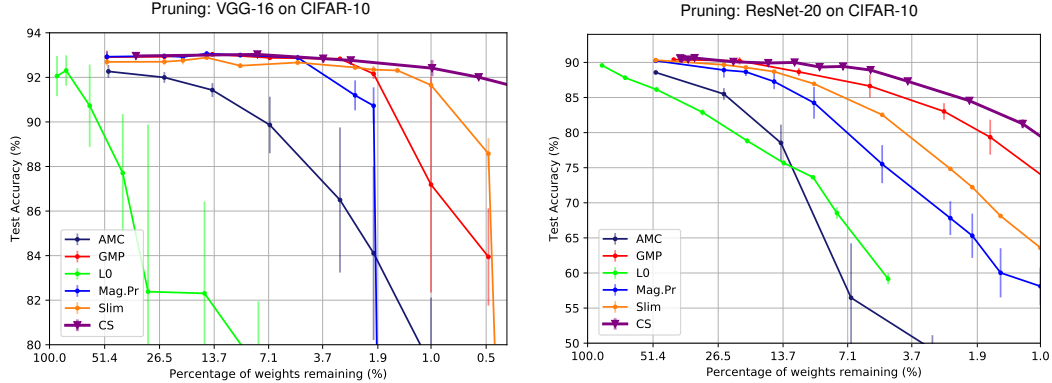

Figure 2: Performance of different methods when performing one-shot pruning on VGG-16 and ResNet-20, measured in terms of test accuracy and sparsity of produced subnetworks after fine-tuning.

Table 2: Sparsity (%) of the sparsest subnetwork within 2% test accuracy of the original dense model, for different pruning methods on CIFAR.

|  | [12] | AMC | MP | GMP | NetSlim | CS |
|---|---|---|---|---|---|---|
| VGG-16 | 18.2% | 86.0% | 97.5% | 98.0% | 99.0% | **99.6%** |
| ResNet-20 | 13.6% | 50.0% | 80.0% | 86.0% | 85.0% | **94.4%** |

GMP [4], and Network Slimming (Slim) [30], along with the $\ell_0$-based method of Louizos *et al.* [12] (referred to as "$\ell_0$"), which, in contrast to ours, adopts a stochastic approximation for $\ell_0$ regularization.

We train VGG-16 and ResNet-20 on CIFAR-10 for 200 epochs, with a initial learning rate of $0.1$ which is decayed by a factor of $10$ at epochs 80 and 120. The subnetwork is produced at epoch 160 and is then fine-tuned for 40 extra epochs with a learning rate of $0.001$. More specifically, at epoch 160 the subnetwork structured is fixed: AMC, MP, GMP and Slim zero-out elements in the binary matrix $m$ for the last time, while CS fixes $m = H(s)$ and stops training of the mask parameters $s$.

Adopting the inference behavior suggested in Louizos *et al.* [12] for $\ell_0$, *i.e.,* using the expected value of the uniform distribution to generate hard concrete samples, leads to poor results, including accuracy akin to random guessing at sparsity above $90\%$; this is also reported in Gale *et al.* [26]. Instead, at epoch 160, we sample different masks and commit to the one that performs the best – this strategy results in drastic improvements at high sparsity levels. This suggests that the gap between training and inference behavior introduced by stochastic approaches can be an obstacle. Although our modification improves results for $\ell_0$, the method still performs poorly compared to alternatives.

Moreover, some methods required modifications as they were originally designed to perform structured pruning. For AMC, Slim, and $\ell_0$ we replace a filter-wise mask by one that acts over weights. Since Network Slimming relies on the filter-wise scaling factors of batch norm, we introduce weight-wise scaling factors which are trained jointly with the weights. We observe that applying both $\ell_1$ and $\ell_2$ regularization to the scaling parameters, as done by Liu *et al.* [30], yields inferior performance, which we attribute to over-regularization. A grid search over the penalty of each norm regularizer shows that only applying $\ell_1$ regularization with a strength of $\lambda_1 = 10^{-5}$ for ResNet-20 and $\lambda_1 = 10^{-6}$ for VGG-16 improves results.

Figure 2 displays one-shot pruning results. On VGG, only CS and Slim successfully prune over $98\%$ of the weights without severely degrading the performance of the model, while on ResNet the best results are achieved by CS and GMP. Table 2 shows the percentage of weights that each method can remove while maintaining a performance within $2\%$ of the original, dense model. CS is capable of removing significantly more parameters than all competing methods on both networks: on ResNet-20, the pruned network found by CS contains $60\%$ less parameters than the one found by GMP, when counting prunable parameters only. CS not only offers significantly superior performance compared to the prior $\ell_0$-based method of Louizos *et al.* [12], but also comfortably outperforms all other methods, providing a new state-of-the-art for network pruning.

### 5.3 Residual Networks on ImageNet

We perform pruning and ticket search for ResNet-50 trained on ImageNet [24]. Following Frankle *et al.* [19], we train the network with SGD for 90 epochs, with an initial learning rate of 0.1 that is decayed by a factor of 10 at epochs 30 and 60. We use a batch size of 256 distributed across 4 GPUs and a weight decay of 0.0001. We run CS for a single round due to the high computational cost of training ResNet-50 on ImageNet. Once the round is complete, we evaluate the performance of the produced subnetwork when fine-tuned (pruning) or re-trained from an early iterate (ticket search).

We run CS with $s^{(0)} \in \{0.0, -0.01, -0.02, -0.03, -0.05\}$ yielding 5 subnetworks with varying sparsity levels. Table 3 summarizes the results achieved by CS, IMP, and current state-of-the-art pruning methods GMP [4], STR [10], and DNW [9]. A $^\dagger$ superscript denotes results of a re-trained, rather than fine-tuned, subnetwork. Differences in each technique's methodology – for example, the adopted learning rate schedule and number of epochs – complicate the comparison.

CS produces subnetworks that, when re-trained, outperform the ones found by IMP by a comfortable margin (compare CS$^\dagger$ and IMP$^\dagger$). Moreover, when evaluated as a pruning method, CS outperforms all competing approaches, especially in the high-sparsity regime. Therefore, our method provides state-of-the-art results whether the network is fine-tuned (pruning) or re-trained (ticket search).

Table 3: Performance of found ResNet-50 subnetworks on ImageNet.

| Method | Top-1 Acc. | Sparsity |
|---|---|---|
| GMP | 73.9% | 90.0% |
| DNW | 74.0% | 90.0% |
| STR | 74.3% | 90.2% |
| IMP$^\dagger$ | 73.6% | 90.0% |
| CS$^\dagger$ | **75.5%** | 91.8% |
| GMP | 70.6% | 95.0% |
| DNW | 68.3% | 95.0% |
| STR | 70.4% | 95.0% |
| CS | **72.4%** | 95.3% |
| IMP$^\dagger$ | 69.2% | 95.0% |
| CS$^\dagger$ | **71.1%** | 95.3% |
| STR | 67.2% | 96.5% |
| CS | **71.4%** | 97.1% |
| CS$^\dagger$ | 69.6% | 97.1% |
| GMP | 57.9% | 98.0% |
| DNW | 58.2% | 98.0% |
| STR | 61.5% | 98.5% |
| CS | **70.0%** | 98.0% |
| CS$^\dagger$ | 67.9% | 98.0% |
| GMP | 44.8% | 99.0% |
| STR | 54.8% | 98.8% |
| CS | **66.8%** | 98.9% |
| CS$^\dagger$ | 64.9% | 98.9% |

## 6 Discussion

With Frankle and Carbin [13], we now realize that sparse subnetworks can indeed be successfully trained from scratch or an early iterate, putting in question whether overparameterization is required for proper optimization of neural networks. Such subnetworks can potentially decrease the required resources for training deep networks, as they are shown to transfer between different, but similar, tasks [16, 17].

The *search* for winning tickets is a poorly explored problem, with, prior to our work, Iterative Magnitude Pruning [13] standing as the only algorithm suited for this task. It is unclear whether IMP's key ingredients – post-training magnitude pruning and parameter rewinding – are the correct choices. Here, we approach the problem of finding sparse subnetworks as an $\ell_0$-regularized optimization problem, which we approximate through a smooth relaxation of the step function.

Our proposed algorithm, Continuous Sparsification, relies on a deterministic approximation of $\ell_0$ regularization, removes parameters automatically and continuously during training, and can be fully described by the optimization framework. We show empirically that, indeed, post-training pruning might not be the most sensible choice for ticket search, raising questions on how the search for tickets differs from standard network compression. In tasks such as pruning VGG and finding winning tickets in ResNets, our method offers improvements in terms of ticket search and resulting sparsity – we can sparsify VGG to extreme levels, and speed up ticket search using an efficiently parallelizable framework. We hope to further motivate the problem of *quickly* finding tickets in complex networks, as the task might be highly relevant to transfer learning and mobile applications.

At the same time, Continuous Sparsification serves as a practical network pruning method, outperforming modern competitors as measured by accuracy and sparsity of produced subnetworks. Continuous Sparsification's principled formulation has the potential to open new avenues for research into neural network optimization and architecture search.

## Broader Impact

Training deep neural networks usually requires significant computational resources. Additional efforts are often needed to prune trained networks to enable efficient inference – for example, in mobile applications which may be both power and compute constrained. Our work presents a new technique via which to sparsify networks, and contributes to the analysis of the recently discovered scientific phenomenon of re-trainable subnetworks (tickets). These contributions might open new pathways towards reducing the computational resources required for deep learning, thereby having a potentially wide-ranging practical impact across the field.

## Acknowledgments and Disclosure of Funding

We thank the anonymous reviewers for providing extensive and extremely valuable feedback on earlier drafts of this work.

The University of Chicago CERES Center contributed to the financial support of Pedro Savarese. The authors have no competing interests.

## Footnotes

[1]Code available at https://github.com/lolemacs/continuous-sparsification

[2]In experiments, we observed that a final temperature of 500 is sufficient for iterates of $s$ when training with SGD using 32-bit precision. The required temperature is likely to depend on how $s$ is represented numerically, as our implementation relies on numerical imprecision rather than (alternatively) clamping after some threshold.

[3]We used the same network as Frankle and Carbin [13] and Frankle *et al.* [19], who refer to it as ResNet-18.

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
