[Supplementary Material]

# Appendix

## A   Hyperparameter Analysis

### A.1   Continuous Sparsification

In this section, we study how the hyperparameters of Continuous Sparsification affect its behavior in terms of sparsity and performance of the produced tickets. More specifically, we consider the following hyperparameters:

- Final temperature $\beta^{(T)}$: the final value for $\beta$, which controls how close to the original $\ell_0$-regularized problem the proxy objective $L_\beta(w, s)$ is.
- $\ell_1$ penalty $\lambda$: the strength of the $\ell_1$ regularization applied to the soft mask $\sigma(\beta s)$, which promotes sparsity.
- Mask initial value $s^{(0)}$: the value used to initialize all components of the soft mask $m = \sigma(\beta s)$, where smaller values promote sparsity.

Our setup is as follows. To analyze how each of the 3 hyperparameters impact the performance of Continuous Sparsification, we train ResNet-20 on CIFAR-10 (following the same protocol from Section 5.1), varying one hyperparameter while keeping the other two fixed. To capture how hyperparameters interact with each other, we repeat the described experiment with different settings for the fixed hyperparameters.

Since different hyperparameter settings naturally yield vastly distinct sparsity and performance for the found tickets, we report relative changes in accuracy and in sparsity.

In Figure 3, we vary $\lambda$ between 0 and $10^{-8}$ for three different $(s^{(0)}, \beta^{(T)})$ settings: $(s^{(0)} = -0.2, \beta^{(T)} = 100)$, $(s^{(0)} = 0.05, \beta^{(T)} = 200)$, and $(s^{(0)} = -0.3, \beta^{(T)} = 100)$. As we can see, there is little impact on either the performance or the sparsity of the found ticket, except for the case where $s^{(0)} = 0.05$ and $\beta^{(T)} = 200$, for which $\lambda = 10^{-8}$ yields slightly increased sparsity.

Figure 3: Impact on relative test accuracy and sparsity of tickets found in a ResNet-20 trained on CIFAR-10, for different values of $\lambda$ and fixed settings for $\beta^{(T)}$ and $s^{(0)}$.

Next, we consider the fixed settings $(s^{(0)} = -0.2, \lambda = 10^{-10})$, $(s^{(0)} = 0.05, \lambda = 10^{-12})$, $(s^{(0)} = -0.3, \lambda = 10^{-8})$, and proceed to vary the final inverse temperature $\beta^{(T)}$ between 50 and 200. Figure 4 shows the results: in all cases, a larger $\beta$ of 200 yields better accuracy. However, it decreases sparsity compared to smaller temperature values for the settings $(s^{(0)} = -0.2, \lambda = 10^{-10})$ and $(s^{(0)} = -0.3, \lambda = 10^{-8})$, while at the same time increasing sparsity for $(s^{(0)} = 0.05, \lambda = 10^{-12})$. While larger $\beta$ appear beneficial and might suggest that even higher values should be used, note that, the larger $\beta^{(T)}$ is, the earlier in training the gradients of $s$ will vanish, at which point training of the mask will stop. Since the performance for temperatures between 100 and 200 does not change

Figure 4: Impact on relative test accuracy and sparsity of tickets found in a ResNet-20 trained on CIFAR-10, for different values of $\beta^{(T)}$ and fixed settings for $\lambda$ and $s^{(0)}$.

Figure 5: Impact on relative test accuracy and sparsity of tickets found in a ResNet-20 trained on CIFAR-10, for different values of $s^{(0)}$ and fixed settings for $\beta^{(T)}$ and $\lambda$.

significantly, we recommend values around 150 or 200 when either pruning or performing ticket search.

Lastly, we vary the initial mask value $s^{(0)}$ between $-0.3$ and $+0.3$, with hyperpameter settings $(\beta^{(T)} = 100, \lambda = 10^{-10})$, $(\beta^{(T)} = 200, \lambda = 10^{-12})$, and $(\beta^{(T)} = 100, \lambda = 10^{-8})$. Results are given in Figure 5: unlike the exploration on $\lambda$ and $\beta^{(T)}$, we can see that $s^{(0)}$ has a strong and consistent effect on the sparsity of the found tickets. For this reason, we suggest proper tuning of $s^{(0)}$ when the goal is to achieve a specific sparsity value. Since the percentage of remaining weights is monotonically increasing with $s^{(0)}$, we can employ search strategies over values for $s^{(0)}$ to achieve pre-defined desired sparsity levels (*e.g.*, binary search). In terms of performance, lower values for $s^{(0)}$ naturally lead to performance degradation, since sparsity quickly increases as $s^{(0)}$ becomes more negative.

Figure 6: Performance of tickets found by Iterative Magnitude Pruning in a ResNet-20 trained on CIFAR-10, for different pruning rates.

## A.2 Iterative Magnitude Pruning

Here, we assess whether the running time of Iterative Magnitude Pruning can be improved by increasing the amount of parameters pruned at each iteration. The goal of this experiment is to evaluate if better tickets (both in terms of performance and sparsity) can be produced by more aggressive pruning strategies.

Following the same setup as the previous section, we train ResNet-20 on CIFAR-10. We run IMP for 30 iterations, performing global pruning with different pruning rates at the end of each iteration. Figure 6 shows that the performance of tickets found by IMP decays when the pruning rate is increased to $40\%$. In particular, the final performance of found tickets is mostly monotonically decreasing with the number of remaining parameters, suggesting that, in order to find tickets which outperform the original network, IMP is not compatible with more aggressive pruning rates.

## B  Iterative Stochastic Sparsification

---

**Algorithm 3** Iterative Stochastic Sparsification (inspired by [18])

**Input:** Mask init $s^{(0)}$, penalty $\lambda$, number of rounds $R$, iterations per round $T$, rewind point $k$

  1: Initialize $w \sim \mathcal{D}$, $s \leftarrow s^{(0)}, r \leftarrow 1$
  2: Minimize $\mathbb{E}_{m \sim \mathrm{Ber}(\sigma(s))} \left[ L(f(\,\cdot\,; m \odot w)) \right] + \lambda \left\| \sigma(s) \right\|_1$ for $T$ iterations, producing $w^{(T)}$ and $s^{(T)}$
  3: If $r = R$, sample $m \sim \mathrm{Ber}(\sigma(s^{(T)}))$ and output $f(\,\cdot\,; m \odot w^{(k)})$
  4: Otherwise, set $w \leftarrow w^{(k)}$, $s \leftarrow -\infty$ for components of $s$ where $s^{(T)} < s^{(0)}$, $r \leftarrow r + 1$ and go back to step 2, starting a new round

---

Besides comparing our proposed method to Iterative Magnitude Pruning (Algorithm 1), we also design a baseline method, Iterative Stochastic Sparsification (ISS, Algorithm 3), motivated by the procedure in Zhou *et al.* [18] to find a binary mask $m$ with gradient descent in an end-to-end fashion. More specifically, ISS uses a stochastic re-parameterization $m \sim \mathrm{Bernoulli}(\sigma(s))$ with $s \in \mathbb{R}^d$, and

Figure 7: Learning a binary mask with weights frozen at initialization with Stochastic Sparsification (SS, Algorithm 3 with one iteration) and Continuous Sparsification (CS), on a 6-layer CNN on CIFAR-10. **Left:** Training curves with hyperparameters for which masks learned by SS and CS were both approximately $50\%$ sparse. CS learns the mask significantly faster while attaining similar early-stop performance. **Right:** Sparsity and test accuracy of masks learned with different settings for SS and CS: our method learns sparser masks while maintaining test performance, while SS is unable to successfully learn masks with over $50\%$ sparsity.

trains $w$ and $s$ jointly with gradient descent and the straight-through estimator [27]. Note that the method is also similar to the one proposed by Srinivas *et al.* [11] to prune networks. The goal of this baseline and comparisons is to evaluate whether the deterministic nature of CS's re-parameterization is advantageous when performing sparsification through optimization methods.

When run for multiple iterations, all components of the mask parameters $s$ which have decreased in value from initialization are set to $-\infty$, such that the corresponding weight is permanently removed from the network. While this might look arbitrary, we observe empirically that ISS was unable to remove weights quickly without this step unless $\lambda$ was chosen to be large – in which case the model's performance decreases in exchange for sparsity.

We also observe that the mask parameters $s$ require different settings in terms of optimization to be successfully trained. In particular, Zhou *et al.* [18] use SGD with a learning rate of 100 when training $s$, which is orders of magnitude larger than the one used when training CNNs. Our observations are similar, in that typical learning rates on the order of 0.1 cause $s$ to be barely updated during training, which is likely a side-effect of using gradient estimators to obtain update directions for $s$. The following sections present experiments that compare IMP, CS and ISS on ticket search tasks.

## C   Supermask Search on a 6-layer CNN

We train a neural network with 6 convolutional layers on the CIFAR-10 dataset [23], following Frankle and Carbin [13]. The network consists of three blocks of two resolution-preserving convolutional layers followed by $2 \times 2$ max-pooling, where convolutions in each block have $64, 128$, and $256$ channels, a $3 \times 3$ kernel, and are immediately followed by ReLU activations. The blocks are followed by fully-connected layers with $256, 256$, and $10$ neurons, with ReLUs in between. The network is trained with Adam [31] with a learning rate of $0.0003$ and a batch size of $60$.

As a first baseline, we consider the task of learning a "supermask" [18]: a binary mask $m$ that aims to maximize the performance of a network with randomly initialized weights once the mask is applied. This task is equivalent to pruning a randomly-initialized network since weights are neither updated during the search for the supermask, nor for the comparison between different methods.

We only compare ISS and CS for this specific experiment: the reason not to consider IMP is that, since the network weights are kept at their initialization values, IMP amounts to removing the weights whose initialization were the smallest. Hence, we compare ISS and CS, where each method is run for a single round composed of 100 epochs. In this case, where it is run for a single round, ISS is equivalent to the algorithm proposed in Zhou *et al.* [18] to learn a supermask, referred here as simply Stochastic Sparsification (SS). We control the sparsity of the learned masks by varying $s^{(0)}$ and $\lambda$. All

Figure 8: Accuracy and sparsity of tickets produced by IMP, ISS and CS after re-training, starting from initialization. Tickets are extracted from a Conv-6 network trained on CIFAR-10. Purple curves show individual runs of CS, while green curve connects tickets produced after 4 rounds of CS with varying $s^{(0)}$. Blue and red curves show performance and sparsity of tickets produced by IMP and ISS, respectively. Error bars depict variance across 3 runs.

parameters are trained using Adam and a learning rate of $3 \times 10^{-4}$, excluding the mask parameters $s$ for SS, for which we adopted SGD with a learning rate of $100$ – following Zhou *et al.* [18] and the discussion in the previous section.

Figure 7 presents results: CS is capable of finding high performing sparse supermasks (*i.e.,* $25\%$ or less remaining weights while yielding $75\%$ test accuracy), while SS fails at finding competitive supermasks for sparsity levels above $50\%$. Moreover, CS makes faster progress in training, suggesting that not relying on gradient estimators indeed results in better optimization and faster progress when measured in epochs or parameter updates.

## D   Ticket Search on a 6-layer CNN

In what follows we compare IMP, ISS and CS in the task of finding winning tickets on the Conv-6 architecture used in the supermask experiments in Appendix C. The goal of these experiments is to assess how our deterministic re-parameterization compares to the common stochastic approximations to $\ell_0$-regularization [11, 12, 18]. Therefore, we run CS **with weight rewinding** between rounds, so that we remove any advantages that might be caused by not performing weight rewinding – in this case, we better isolate the effects caused by our re-parameterization. Following Frankle and Carbin [13], we re-train the produced tickets from their values at initialization (*i.e.,* $k = 0$ on each algorithm).

We run IMP and ISS for a total of 30 rounds, each consisting of 40 epochs. Parameters are trained with Adam [31] with a learning rate of $3 \times 10^{-4}$, following Frankle and Carbin [13]. For IMP, we use pruning rates of $15\%/20\%$ for convolutional/dense layers. We initialize the Bernoulli parameters of ISS with $s^{(0)} = \vec{1}$, and train them with SGD and a learning rate of $20$, along with a $\ell_1$ regularization of $\lambda = 10^{-8}$. For CS, we train both the weights and the mask with Adam and a learning rate of $3 \times 10^{-4}$. Each run of CS is limited to $4$ rounds, and we perform a total of 16 runs, each with a

different value for the mask initialization $s^{(0)}$, from $-0.2$ up to $0.1$. Runs are repeated with 3 different random seeds so that error bars can be computed.

Figure 8 presents tickets produced by each method, measured by their sparsity and test accuracy when trained from scratch. Even when performing weight rewinding, CS produces tickets that are significantly superior than the ones found by ISS, both in terms of sparsity and test accuracy, showing that our deterministic re-parameterization is fundamental to finding winning tickets.

## E    Additional Plots for Ticket Search Experiments

In Section 5.1, we compare IMP and CS in the task of performing ticket search for ResNet-20 trained on CIFAR-10, where CS was run with 11 different values for $s^{(0)}$ in order to produce tickets with diverse sparsity levels, each run consisting of 5 rounds.

Figures 9 and 10 contain the training curves for each of the 11 settings of $s^{(0)}$ that produce tickets presented in Figure 1 (left). Purple curves show the performance and sparsity of tickets produced after each of the 5 rounds. The accuracy for each ticket is computed by re-training from early-training weights (epoch 2). For each setting, we also execute IMP with a pruning rate per round matching CS, which is presented a blue curve – note that these runs of IMP are different than the ones in Section 5.1, where IMP had a fixed and pre-defined pruning ratio of $20\%$ per round.

The plots show that CS not only adjusts the per-round pruning ratio automatically, but it is also superior in terms of what parameters are removed from the network. The bottom right plot of Figure 10 shows curves connecting tickets that are presented in all other plots of Figure 9 and 10 (left), where we can see that CS produces superior tickets even when IMP adopts a dynamic pruning ratio that matches the one of CS at each round.

Figure 9: Accuracy and sparsity of tickets produced by IMP and CS after re-training, starting from weights of epoch 2. Tickets were extracted from a ResNet-20 trained on CIFAR-10. Each plot corresponds to different value for the mask initialization $s^{(0)}$ of CS, ranging from $0.3$ to $0.0$, with IMP adopting the same pruning rate per round. Ticket performance is given by purple curves when produced by CS, while blue shows performance of IMP and continued IMP (IMP without weight rewinding between rounds).

Figure 10: Accuracy and sparsity of tickets produced by IMP and CS after re-training, starting from weights of epoch 2. Tickets were extracted from a ResNet-20 trained on CIFAR-10. Each plot corresponds to different value for the mask initialization $s^{(0)}$ of CS, ranging from $-0.03$ to $-0.3$, with IMP adopting the same pruning rate per round. Ticket performance is given by purple curves when produced by CS, while blue shows performance of IMP and continued IMP (IMP without weight rewinding between rounds). The bottom right plot shows performance of tickets produced during runs corresponding to all other plots in Figures 9 and 10.

## F  Sequential Search with Continuous Sparsification

There might be cases where the goal is either to find a ticket with a specific sparsity value or to produce a set of tickets with varying sparsity levels in a single run – tasks that can be naturally performed with a single run of Iterative Magnitude Pruning. However, Continuous Sparsification has no explicit mechanism to control the sparsity of the produced tickets, and, as shown in Section 5.1 and Appendix E, CS quickly sparsifies the network in the first few rounds and then roughly maintains the number of parameters during the following rounds until the end of the run. In this scenario, IMP

Figure 11: Accuracy and sparsity of tickets produced by IMP and Sequential CS after re-training, starting from weights of epoch 2. Tickets are extracted from a ResNet-20 trained on CIFAR-10.

has a clear advantage, as a single run suffices to produce tickets with varying, pre-defined sparsity levels.

Here, we present a sequential variant of CS, named Sequential Continuous Sparsification, that removes a fixed fraction of the weights at each round, hence being better suited for the task described above. Unlike IMP, this sequential form of CS removes the weights with lowest mask values $s$ – note the difference from CS, which, given a large enough temperature $\beta$, removes *all* weights whose corresponding mask parameters are negative.

Following the same experimental protocol from Section 5.1, we again perform ticket search on ResNet-20 trained on CIFAR-10. We run Sequential Continuous Sparsification and Iterative Magnitude Pruning for a total of 30 rounds each, and with a pruning rate of $20\%$ per round. Note that unlike the experiments with Continuous Sparsification (the non-sequential form), we perform a single run with $s^{(0)} = 0$, *i.e.,* no hyperparameters are used to control the sparsity of the produced tickets.

Figure 11 shows the performance of tickets produced by Sequential CS and IMP, indicating that CS might be a competitive method in the sequential search setting. Note that the performance of the tickets produced by Sequential CS is considerably inferior to those found by CS (refer to Section 5.1, Figure 1). Although these results are promising, additional experiments would be required to more thoroughly evaluate the potential of Sequential Continuous Sparsification and its comparison to Iterative Magnitude Pruning.

## G   Learned Sparsity Structure

To see how CS differs from magnitude pruning in terms of which layers are more heavily pruned by each method, we force the two to prune VGG to the same sparsity level in a single round. We first run CS with $s^{(0)} = 0$, yielding $94.19\%$ sparsity, and then run IMP with global pruning rate of $94.19\%$, producing a sub-network with the same number of parameters.

Figure 12 shows the final sparsity of blocks consisting of two consecutive convolutional layers (8 blocks total since VGG has 16 convolutional layers). CS applies a pruning rate that is roughly twice as aggressive as IMP to the first blocks. Both methods heavily sparsify the widest layers of VGG

Learned Sparsity Patterns in VGG on CIFAR-10

Figure 12: Sparsity patterns learned by CS and IMP for VGG-16 trained on CIFAR-10 – each block consists of 2 non-overlapping consecutive layers of VGG.

(blocks 5 to 8), while still achieving over $91\%$ test accuracy. More heavily pruning earlier layers in CNNs can offer inference speed benefits: due to the increased spatial size of earlier layers' inputs, each weight is used more times and has a larger contribution in terms of FLOPs.