[Reviews · NeurIPS 2020]

Review 1

Summary and Contributions: This paper proposes using continuous approximation of l_0 regularization to introduce sparsity during the training of a dense network. Experimental results show improved results over baselines, especially at high sparsities.

Strengths: The paper is well written and have strong experimental results.

Weaknesses: - Novelty? Technique proposed reminds https://arxiv.org/pdf/1712.01312.pdf. Since the formulation and main approach is quite similar. It would be nice to include a detailed comparison with this work that shows similarities and differences. - As mentioned in the main text one drawback of the method proposed is that it is not able to match exact sparsities and one needs to play with various hyper-parameters (which is fine). It would be nice to have a comparison though with the version where beta=1 and a threshold is used after training to generate masks. With this the method might not need to rely on numerical imprecision. - I find the motivation a bit confusing. Training sparse networks from scratch is an important problem as pointed out by LTH. However, LTH is not practical. This paper doesn't try to solve this problem as the algorithm needs multiple rounds. Even when it is used with a single round, it requires dense resources. Therefore I find it confusing to motivate the entire work with LTH. It would be more appropriate to motivate CS as pruning algorithm and compare them with SOTA pruning methods (instead of IMP, which is not a good pruning algorithm and it is extremely inefficient). - Authors compare their method with rigged lottery (rigl), which trains sparse networks, (in theory) without any dense resources. I find it unfair to compare a sparse training method with a pruning algorithm as the first one requires much less resources. Therefore, it would be nice to include 5x results or mention this important detail in the text. Additionally I recommend authors using ERK distribution results as CS does global pruning and resulting sparsity distributions are non-uniform.

Correctness: Some minor points: a) `2 A key finding is that the same parameter initialization should be used w` This is not a necessary condition. This is one way, we observe training works fine. I would rephrase the sentence to indicate that. b) `typically perform weight selection and removal once the dense model has been fully trained.` This is not true in recent methods. Most pruning methods now combine training and pruning and can be done in a single run. One example: https://arxiv.org/abs/1710.01878

Clarity: It's mostly well written and clear. Some minor points: - ...found tickets -> tickets found tools to finding small -> tools for finding Unlike concurring works -> unlike [x,y,z] ? does not rewinding weigh -> not rewind CS for Continuous Sparsification defined in page-5. I would recommend defining it earlier and using CS in the rest of the paper. `and CSafter re-training,` -> CS after

Relation to Prior Work: a) L28: `The fact that pruned networks are hard to train from scratch [3] suggests that`: It might be better to cite the original LTH or (https://arxiv.org/abs/1906.10732). I don't think [3] have training from scratch baseline and/or focuses on this problem. b) `With [8], we now realize that sparse sub-networks can indeed be successfully trained from scratch, putting in question whether over-parameterization is required for proper optimization of neural networks.` I understand authors mean the resulting LT initialization works fine, however we still need to do the pruning. Therefore real training from scratch attempted/done by other works: i.e. snip, grasp, set, rigl. It might be better to cite these work here or rephrase the sentence.

Reproducibility: Yes

Additional Feedback: I am willing to update my score if my concerns are addressed during rebuttal. //AFTER REBUTTAL// I like to thank authors for their responses. 1) DNW and STR are pruning methods, in other words they start with dense network and produce sparse results. These comparisons together with the standard magnitude pruning, i.e. Gupta,2017 pruning, would help the reader appreciate the work better. It looks like CS is getting better results. I don't think removing these results makes sense. 2) FLOPs needed for inference along with layer-wise sparsities are important to report. One easy trick for getting better performance at same sparsity is to use non-uniform sparsity distributions like ERK or the ones found by STR. It is very easy to report those and they are very helpful for assessing results. 3) Another interesting ablation would be to use the layer-wise sparsities found by CS and try them with GMP (or other methods that support custom sparsities). This could help authors to further disentangle the effect of regularization and parameter allocation over layers.


Review 2

Summary and Contributions: If the authors sufficiently address my concerns, I will champion this paper for acceptance. This paper proposes a new technique for finding the "matching subnetworks" (subnetworks that could have trained in isolation to full accuracy from an early point in training) at the heart of work on the so-called "lottery ticket hypothesis." The technique, called Continuous Sparsification (CS), learns which weights to prune by maintaining a per-weight parameter whose value is used to eliminate weights. The technique is modification to an existing pruning technique called "l0 regularization" (Louizos et al.). CS finds sparser matching subnetworks than the incumbent method (IMP). When provided with sufficient parallel compute, CS can find find matching subnetworks at all sparsities faster than IMP. I specialize in lottery ticket research, and I consider CS to be a major advance. It allows us to improve our knowledge of the sparsest possible matching subnetworks and it makes it possible for us to efficiently scale up the lottery ticket observations. Experiments that might have taken multiple weeks with IMP (e.g., producing subnetworks at all sparsities on ResNet-50 for ImageNet) now take days (perhaps even less) with CS given sufficient parallel compute. Despite these strengths, the paper is not without major weaknesses in the claims, the technical content, and the writing. There is plenty of room for improvement, and I implore the authors to address my concerns (see the "Weaknesses" and "Clarity" sections below). If the authors acknowledge and address my concerns where requested below (or commit to address any concerns that cannot be addressed in the brief author response window), I will maintain my score and champion this paper for acceptance. If they do not, I will lower my score. UPDATE AFTER AUTHOR RESPONSE ============================== Thank you to the authors for their thoughtful response. I continue to champion this paper for acceptance. All of my concerns received a response in the author feedback (see below). ***Although there is not another phase of review, I look forward to carefully reading the next version of the paper and ensuring that the authors keep their promises on (1), (2), and (3), addressing them in the manner requested below.*** (1) Writing quality in Sections 1 and 2. The authors have promised "a subsection introducing the reader to the nomenclature and precise definitions." ***I look forward to reading the next version of the paper and holding the authors accountable on this point. This section should appear early in the paper (perhaps before any other content save the introduction) so there is no possible confusion.*** (2) Relationship between CS and L0. The authors have thoroughly addressed my concerns on this front. ***Considering that all four reviewers were confused on this point, the authors need to update the camera ready to feature this discussion prominently in place of the brief, hand-wavey comparisons that are currently in the paper.*** This is not just to satisfy me as a reviewer - this is for your own good: it will detract from the impact your paper has if other readers are similarly confused on this point. (3) CS as a pruning technique vs. a technique for retroactively finding matching subnetworks. I appreciate the efforts the authors made on this front during the short author response period. However, I do not think the authors fully appreciate the gravity of this concern. I agree with Reviewer 1: if "ticket search is a strictly more general and harder task than pruning," then there is no excuse for the paper not to feature comparisons to standard pruning methods. ***I believe that this pruning content should be featured with equal prominence to the lottery ticket experiments rather than as an afterthought (as in the current version). I will not stand in the way of acceptance on this point alone, but it is crucial that the authors take this concern seriously. I look forward to seeing this content featured prominently in the next version of the paper.*** (4) CS hyperparameters. I am satisfied on this point. (5) Comparisons to RigL, etc. On this front, I defer to Reviewer 1. Please update the paper according to Reviewer 1's feedback.

Strengths: This section will be short because the strengths are simple: the authors propose an easy-to-understand method that provides important wins for finding sparser matching subnetworks and, in the right circumstances, finding them more efficiently. The paper is has a clear goal and conducts thorough experiments to demonstrate that the proposed technique meets that goal. It doesn't have to be any more complicated than that. It's worth emphasizing that this paper follows all of the empirical best practices. The evaluation metrics for evaluating are clear and well-motivated. Experiments are conducted on many networks, including large-scale settings. Experiments include multiple replicates and error bars. All details and hyperparameters are present, meaning it would be easy to reproduce these results. The paper also does an excellent job acknowledging the limitations of CS. It is completely acceptable that CS isn't the perfect tool for every use case, and the authors should be commended for being upfront about it. This is unusual in deep learning, but it is much easier to trust the claims of strengths in the paper because the authors are willing to admit weaknesses. I request that the authors add a formal "Limitations" section prior to the "Discussion" section to collect together the assorted weaknesses of CS mentioned throughout the paper. That will be an excellent way to make it clear where others can follow up to improve upon this technique and to set a good example for work that is less forthcoming about limitations.

Weaknesses: Weaknesses marked with a *** need to be addressed in the author response. ***I have significant concerns about the quality of the writing in Sections 1 and 2. Please see the "Clarity" section below for more details. ***I am confused about the relationship between CS and the l0 regularization technique (Louizos et al.). The first paragraph in Section 2 discusses the weaknesses of prior stochastic l0 regularization techniques and suggests that CS is better. But the paper never compares CS and l0 to demonstrate whether CS actually improves performance/reliability/usability. What is the motivation for using CS rather than l0? Would l0 have sufficed to produce the same results as CS in Section 4? What is the value proposition for CS over l0 in terms of performance, reliability, or usability? (You should answer this question with concrete numbers; it would be sufficient to show that performance is the same but CS is simpler, easier to implement, or more consistent.) My broader concern is the possibility that CS is a cosmetic modification of l0 for the sake of making it look novel, but that it doesn't improve upon l0 in any way (or even performs worse than l0). To address this concern, the authors should add these comparisons, even in an appendix. If the authors cannot show that CS has concrete advantages over l0 as requested above, that undermines the case for CS, and this becomes a still-publishable but less exciting paper about how the l0 technique of Louizos et al. is a better choice than IMP for finding matching subnetworks. ***I am confused about CS as a pruning technique vs. as a technique for finding matching subnetworks. Any technique that finds matching subnetworks is implicitly a pruning technique; just train the matching subnetworks to completion and you have a pruned network. l0 regularization (Louizos et al.) is pitched as a pruning technique, so I see no reason why CS should not be held to the same standard. Right now, finding matching subnetworks is the focus of the paper and pruning is an afterthought at the very end. There is no excuse not to treat these two goals of CS equally; in short, the paper should include more pruning results, specifically on the same networks and datasets as it uses for finding matching subnetworks. This means comparing to l0 regularization as a pruning technique and comparing to state-of-the-art magnitude pruning algorithms ("AMC" by He et al. https://arxiv.org/abs/1802.03494, "Comparing Rewinding and Fine-Tuning" by Renda et al., and GMP by Gale et al. and Zhu & Gupta). At the very least, these baselines could replace STR, RigL, and DNW in Table 2 (which are unfair comparisons, as I mention below). You should run these techniques yourself so that you can make apples-to-apples comparisons with the same hyperparameters and the same sparsities; it's difficult to compare numbers directly across papers. ***CS appears to have several hyperparameters: the schedule for beta, the value of lambda, and the initialization of s. I wish the paper had spent some time in the main body to (1) acknowledge this fact, (2) briefly explain a bit more about what these hyperparameters do, and (3) to note that hyperparameter search is a drawback of CS rather than deferring this content to an appendix. ***Table 2: It isn't fair or appropriate to compare to RigL, DNW, or STR in Table 2. These are not methods for finding winning tickets; they are of a completely different nature with very different goals and tradeoffs. They are designed to train sparse networks from scratch in order to minimize the cost of training, whereas IMP and CS are expensive ways of retroactively finding subnetworks with these properties. Comparing accuracy alone is insufficient to capture the tradeoffs here. By making these gratuitious and unfair comparisons, you undermine your credibility elsewhere in the paper. I strongly recommend you remove those comparisons. Your paper already has big wins - this isn't a necessary or helpful comparison. Figure 1 is very crowded and hard to read. You should delete the reinit lines and probably also the IMP-C line from this graph. They're less important and will make it easier to read. This is the headline graph for your paper (the one people will show off when they talk about it), so it's worth spending a few hours to make it as clear and easy to read as possible. If you only have three sets of lines, it will be much easier to parse. I also recommend that you make a separate legend that you put below the plots (so the text can be bigger), you make the text bigger on the axis ticks and labels, and you give each graph a heading with the name of the network and dataset. You should also make the lines thicker and use the matplotlib "fillbetween" for the error bars rather than the standard matplotlib error bars. Your goal should be to make it possible for these graphs to fully summarize your key result without the need for much context. It will make things clearer to people who skim the paper and help you to market the paper on blogs/Twitter. Table 2: Please clarify whether you computed your own IMP (12x) results or used the ones from Frankle et al. If you used their data, be sure to note that your hyperparameters and hardware are very different than theirs; this can have a big effect on the performance of pruned networks. The paper never specifies the rewinding iterations used for the networks in the paper. That needs to be made clear in the main text.

Correctness: The authors have done an excellent job with the empirical methodology. They have worked hard to make fair comparisons and to acknowledge the weaknesses of their proposed method (for which they should be commended). My only concern is that the comparisons to RigL, DNW, and STR are unfair and should be removed.

Clarity: ***I have severe concerns about the quality of the writing in Sections 1 and 2. To maintain my support for acceptance, the authors need to express that they understand my specific concerns (especially (1)) about the writing and commit to addressing them in the next revision. (1) Throughout the paper (see my detailed comments below), concepts are used without introducing them or fully explaining them. I happen to be intimately familiar with the topics in this paper, so I could still make sense of everything. However, any reader who doesn't work in precisely this space (i.e., someone who doesn't work on lottery tickets or neural network pruning) will be very confused. If the authors hope for others to understand this paper and adopt their technique, they need to revise the prose to ensure it is comprehensible to readers from beyond this narrow subfield. (2) The paper is very imprecise in many places. Important terms ("ticket" vs. "subnetwork" vs. "winning ticket", "iterates") go undefined, and it is assumed that the reader knows the difference between these terms. The paper does not use these terms precisely. (3) The paper misuses the term "winning ticket." "Winning ticket" was defined by Frankle & Carbin [8] as a subnetwork that can train in isolation from *initialization* to full accuracy. In a later paper, Frankle et al. [9] introduce rewinding, in which subnetworks are found *early in training* rather than at initialization; they refer to subnetworks that can train from this point to full accuracy as "matching subnetworks," NOT winning tickets. The reason they do so is because there is no longer an initialization lottery to win if the network has already undergone some training. On line 104, the authors actively choose to override the "matching subnetwork" terminology coined in the paper that they build on. They should use the proper "matching subnetwork" terminology; otherwise, they are just adding confusion to the literature.

Relation to Prior Work: I am confused about the relation to l0 regularization (Louizos et al) as described in the weaknesses section. To satisfy my concern, the authors will need to show concrete numbers that support a case that CS is an improvement over l0 regularization (rather than a cosmetic tweak for the appearance of novelty). To be clear, even if the authors cannot make this case in a satisfactory way, I think the paper should still be accepted on the basis of the great lottery ticket results; however, in that case, the authors should downplay the emphasis on the novelty of CS.

Reproducibility: Yes

Additional Feedback: These are my detailed notes from reading the paper. All concerns that need addressing in the author response have been mentioned elsewhere. These notes include many small nits that I recommend fixing. Abstract: * Nit: Faster and better is only really true on MNIST. In other settings, the results are merely "comparable." Intro: * Nit: Line 22: Should clarify SOTA performance "in practice." * Nit: I find it frustrating when you use the numbered citations as nouns, e.g., "[9] showed that...". Use the names of the authors or say "It was found that... [9]" * Nit: Characterization of the LTH work [8] focuses on "better performance" and "given less epochs," but the hypothesis only claims that networks match the performance in the same amount of time. In general, pruning seems to yield improvements in accuracy; it's not specific to LTH. Seems like you're advertising the LTH work as doing more than it actually accomplishes. * Line 35: need to define what a subnetwork is. * Line 36: What does it mean to say, "re-training can be done"? In LTH work, it's "that can train in isolation to full accuracy." Also, need to clarify whether this is from init ("winning ticket") or from early in training ("matching subnetwork.") In general, use of terminology here could be clearer. * Lines 39-44: If I weren't already intimately familiar with IMP, I would find this paragraph confusing. Need to explain what IMP is concretely first before you talk about the tradeoffs (in particular, mention weight rewinding). * Line 44: You never mention that IMP uses magnitude pruning prior to this point. Also, not sure that [13] really says magnitude is suboptimal, just that there are better ways of utilizing magnitude information. I don't think you need to trash magnitude pruning to motivate your approach, so it might be best to cut this sentence. * Line 48: You never define the term "ticket" vs. "subnetwork" vs. "winning ticket." Need to be precise with this terminology. If I weren't intimately familiar with this topic, I would be confused. * Line 51: "discrete time intervals" - how does this relate to magnitude pruning previously? You need to be more systematic about what information and terminology you introduce at what times. * Line 52: Need to explain what l0 is before you use assume knowledge of it. * Line 57: "Faster learning" - compared to what? Related Work: * Line 73: It's pretty common to prune throughout training. See Gale et al., 2019 and Zhu & Gupta 2017 * Line 79: Need to introduce the notion of a binary mask for pruning somewhere. Might want to have a preliminaries section before this to introduce all your terminology and notation. * Lines 76-87 are pretty confusing. There's a lot going on there, much of whic his important for introducing your technique in Section 3 anyway. You could gloss over that presentation in Section 2 and push some of the content to Section 3 where the appropriate context has been introduced. * Line 89: What does it mean to be "successfully retrained"? Where do these subnetworks come from? That's the essential part of LTH: that they're at init or early in training. This paragraph needs to emphasize that these subnetworks are found at init. It doesn't mention that anywhere. Other than in the mathematical notation and the algorithm notation, this fact isn't mentioned. It would be easy for a reader to assume that these "subnetworks" are just from standard pruning. I know IMP, but I'm guessing most readers would be confused. * Line 104: You are misusing the term "winning ticket." That only refers to a subnetwork from initialization. The broader term "matching subnetwork" refers to subnetworks with these properties from early in training. You need a good reason to justify why you use a method from [9] (rewinding) but explicitly reject the terminology from [9]. Section 3.1 * Line 113: You need to re-orient the reader after the related work. Add a sentence here reminding the reader that you're trying to find a better (be explicit about better in what ways) method for finding matching subnetworks. * Line 113-115: Why are these necessarily advantages? You don't need to say this to motivate your work - just show that it's better. The explanations you pose may not be the reason it's better, so - unless you have evidence this is the case - it's better not to make sweeping generalizations at all. * Line 115: Need to clarify that l0 is usually stochastic. * Does your method work any better than l0 regularization for finding matching subnetworks? I'm confused as to whether you're advertising an improvement over l0, an improvement over IMP, or both. The claims in 113-122 are not enough to justify CS over l0 without actual performance numbers showing there is no degradation from using CS over l0. * Algorithms 1 and 2: Need to introduce the notation (f(.; w), m, element-wise product, etc.) before you use it. This notation is obvious to me because I'm familiar with the prior work in this area; it won't be so for most readers. * Lines 137-140: This paragraph is unnecessary and just adds confusion. At this point, just tell me the method and use the extra space to better explain the content from earlier in the paper. This paragraph (and really, this section) is a nexample of the "Mathiness" troubling trend in machine learning (Lipton & Steinhardt): "use of mathematics which obfuscates or impresses rather than clarifies" * Line 159: "every negative component of s will go to 0" - isn't that sigmoid(beta * s) will go to 0? * It seems like this method has a lot of hyperparameters. Section 3.2 * Thank you for clarifying that CS does not involve rewinding. Section 4 * I enormously appreciate the clarity on the evaluation metrics. * I enormously appreciate the clarity on the tradeoffs with using CS with respect to hyperparameters. One suggestion: as a third metric, you should add the cost of producing tickets at a specific sparsity or at the most extreme matching sparsity. It's completely fine if you don't beat IMP on this front - I still support acceptance regardless. It just makes clear to a reader that this is another intended use case, but is not one where CS is the best choice. * Lines 201-204: You gloss over the role of the hyperparameters (and any difficulty in finding them), but this is important and merits some further discussion. * I appreciate all the additional baselines in the supplement, and that you chose to move the supermask comparisons to the supplement (which emphasizes your main results in the main body). * I appeciate footnote 3 clarifying the mistake in [8]. * Line 215: The setup in 8 for ResNet-20 and VGG-16 isn't ideal. If you re-do these experiments for the next version, I recommend you use the hyperparameters from [9], which are available at https://github.com/facebookresearch/open_lth * For IMP, do you use layerwise pruning or global pruning? I assume it's global, but it's worth clarifying. * Figure 1 is very crowded and hard to read. See the "weaknesses" section for ways to make it better. * It's annoying that Figure 1 isn't on the same page as it is introduced in the text. * Line 235: [9] does perform weight rewinding between rounds. Please fix this (and sorry if that wasn't clear in [9]). * Lines 236-241: This is great. By being clear about your evaluation metrics, you have made it possible to demonstrate indisputable wins. * Lines 242-248: This should be a third evaluation metric that you present in Table 1. It's completely fine if you don't beat IMP on everything. * I don't think it's appropriate to compare to STR, RigL, DNW, etc. See weaknesses. * Pruning results: why isn't this a bigger part of the paper? CS could just as easily be seen as a pruning method, so it's strange that you focus on finding winning tickets rather than pruning. Beating magnitude pruning isn't easy (see Blalock et al. 2020), so that's a big win. * Move Figure 2 (left) and Section 4.4 to an appendix. They're not very important or relevant to the main claims.


Review 3

Summary and Contributions: This article is about the “Lottery Ticket Hypothesis”. The authors propose a new method, “Continuous Sparsification” to find “winning tickets” inspired by one of the most recent works of Frankle et al. (Stabilizing the lottery ticket hypothesis, 2019). The authors have followed the clues provided by Frankle et al., who suggested to improve their (crude) pruning technique. Therefore, the authors have decided to implement a pruning technique based on a l_0 penalty. They show experimentally that their method leads to sparser and more accurate tickets (i.e. subnetworks which are reinitialized to their initial value, then retrained).

Strengths: This article is the continuation of researches about a hot topic (that is the Lottery Ticket Hypothesis). It is then relevant for the deep learning community, especially for research about overparameterization and pruning. As awaited by preceding works, this article provides some results about more sophisticated pruning techniques within the context of Lottery Ticket Hypothesis. Moreover, this new technique leads to sparser and more accurate neural networks.

Weaknesses: This paper presents a new method to find “lottery tickets”, but also a pruning technique. The reader might expect a comparison with very similar pruning techniques, at least in the appendix. For instance, “Learning efficient convolutional networks through network slimming”, Liu 2017 is close to the proposed technique (since it is also based on a penalization of “masks”). I wonder how such existing techniques compare to CS. EDIT: I have read authors' feedback, and I am satisfied with it. The authors have plotted the required comparisons with other pruning techniques.

Correctness: Every step of the algorithm is well justified and referenced (when the authors have reused existing techniques). The article comes with an appendix with several additional experiments, which shows the influence of hyperparameters which are crucial for pruning (weight decay, final “temperature” beta_T, and initial mask value s_0). This is valuable, since it helps the reader to understand how should be run his/her own hyperparameter search, and how many setups should be tested. Besides, the code has been provided, which is helpful to understand exactly what is done. Note: I don’t have tested it. The comparison with the preceding technique seems fair to me.

Clarity: There is no major issue about the way the paper is written.

Relation to Prior Work: I have already explained that this paper is explicitly in line with preceding works about the “Lottery Ticket Hypothesis” (LTH).

Reproducibility: Yes

Additional Feedback: From my point of view, this work leads to one question: in order to find the best “lottery ticket”, what is the most adapted pruning algorithm? Is there some criterion a pruning technique should fulfill in order to be efficient in a LTH context? This question seems important to me, since there exists a huge variety of pruning techniques, and we have to select the best one according to a “novel” criterion, that is: find the best subnetwork to re-train.


Review 4

Summary and Contributions: The authors propose a novel method called Continuous Sparsification (CS) to find the winning lottery ticket. Comparing with traditional Iterative Magnitude Pruning, CS is able to continuously prune parameters during training without multiple re-trainings, which are computationally expensive. The paper empirically validates that CS finds winning tickets up to 5 times faster than IMP, in terms of number of training epochs.

Strengths: 1) Authors evaluate CS on multiple models and datasets including ImageNet, which shows CS outperforms IMP with higher sparsity in terms of test prediction accuracy. 2) CS is more efficient than IMP under parallel computing settings though less efficient if run sequentially. Comparing with IMP, CS yields better performance when doing on-shot pruning (reduce computational cost) but requires hyper-parameter tuning (increase cost). Clearly such a trade-off can be well-balanced under a parallel computing setting which is useful in practice. In addition, the paper claims that the hyperparameters for CS only require limited tuning, which further demonstrates its computational efficiency.

Weaknesses: I have a concern about the paper's novelty. From my understanding present work mostly combines two existed ideas: 1) training sparser networks via approximating an intractable l0-regularization penalty and 2) finding lottery ticket hypothesis under a framework similar to IMP. For the first part, the authors didn't clearly differentiate the present l0-approximation method from previous works. For the second part, the authors did mention that the rewinding for weights is related (to early-stage instead of original initialization) but unfortunately I think it is just a minor change. I believe the novelty of this work depends on the part of l0-approximation and it would be great if the authors can clarify it further. EDIT: Thanks for the authors' response. It clearly alleviates my concern of novelty of l0 approximation. Moreover, other reviews give me a better understanding of this paper. I would like to increase my score and champion the paper for acceptance.

Correctness: The claims and method are correct, as well as methodology.

Clarity: The paper is mostly well written.

Relation to Prior Work: Authors didn't clearly discuss this work differs from previous work on approximating an l0-regularization for training sparse network (section 2.1).

Reproducibility: Yes

Additional Feedback:

[Author Response · NeurIPS 2020]

We thank the reviewers for the feedback and suggestions.

**R1,R2,R3,R4 – Comparison to Louizos *et al.* [1]:** Our approach to approximate the $\ell_0$ objective is fundamentally different from prior work. Louizos *et al.* use tools from variational inference and stochastic optimization (see Appendix A of [1]), while we rely on continuation methods to construct a *deterministic* approximation. A stochastic approximation results in different sets of weights used in each forward pass: refer to Section 5.1 of Gale *et al.* [4], which reports that it failed to sparsify a Wide ResNet "across hundreds of experiments" without resulting in "test set performance akin to random guessing". Refer to the included plot for a comparison with the method of Louizos *et al.* on one-shot pruning a ResNet/VGG: note that we were only able to achieve these results with modifications to the original method, including fixing a stochastically-optimized mask prior to fine-tuning, without which the method fails to yield over $20\%$ sparsity while maintaining performance that is not akin to random guessing. The need to choose a fixed, binary mask sample makes the method even less suited for ticket search. CS relies on a deterministic re-parameterization, avoiding gradient estimators altogether, while also having clear behavior at test time since a single mask is learned.

| Method | $\ell_0$ | Mag.Prune | GMP | CS |
|---|---|---|---|---|
| VGG | 32.0% | 97.5% | 98.0% | 99.6% |
| ResNet | 31.8% | 75.5% | 86.0% | 89.3% |

Highest sparsity achieved by each method while being within $2\%$ (relative) performance of the dense model. Same protocol as in Section 4.3 (one-shot pruning on CIFAR-10).

**R1,R2 – Comparisons:** We will remove RigL, STR, and DNW from our Table 2 to avoid confusion, while still discussing the methods, pointing out differences in goals and methodology. L257 mentions that comparing methods with different methodologies is complicated, but we will remove entries to avoid confusion.

**R1 – Matching pre-defined sparsities:** Appendix F presents a variant of CS that is able to match exact sparsities by defining thresholds to binarize the mask values. For that experiment, we also tried having a fixed $\beta$=1 during training, which led to significantly worse results. **Motivation and LTH:** Ticket search is a strictly more general and harder task than pruning: every ticket search method can be used for pruning, but not every pruning method can perform ticket search (if the learned parameter-wise masks are not binary, then applying it to initial weights would change their magnitude, hence the resulting network would not be a 'ticket'). Moreover, ticket search provides insights into the behavior of neural networks, and is an invaluable tool to study how overparameterization affects optimization dynamics of deep networks. By making ticket search faster and parallelizable, we open doors to large-scale empirical studies on training dynamics of sparse networks. Finally, winning tickets have valuable applications other than just parameter efficiency – for example transfer learning (refer to [10,11,12] of our paper), where ticket search can be performed on small datasets and produced tickets can be successfully re-trained on larger tasks. **More pruning results:** IMP is actually a sensible pruning method: refer to Table 2. Also refer to Renda *et al.* [2] below, which proposes a method that is similar to IMP-Continued and is comparable to the state-of-the-art. The included plot presents comparison to additional methods, and we will extend it further in the camera-ready version.

**R2 – Writing:** We will use space from the extra page given to the camera-ready version to add a subsection introducing the reader to the nomenclature and precise definitions (in the same vein as how we define the Sparsest Matching Ticket and Best Performing Ticket in Section 4). We will also refer to sub-networks that are trained from early iterates as 'matching subnetworks' throughout the paper. **CS as pruning method:** We are currently collecting additional results with recent pruning methods like AMC, GMP (see plot above), and the method in Renda *et al.* [2]. We will add these to the ImageNet results in Table 2, and given space from the extra page for the camera-ready version, add pruning experiments with methods run for multiple rounds, where we can include iterative methods like Renda *et al.*'s approach. **Discussion on hyperparams:** We will use space from the extra page to add to the main text a more extensive discussion on the role of each hyperparameter and how it affects CS, while still keeping full details in the Appendix. Thanks for the detailed review and the list of suggestions; we commit to incorporate all the suggested changes to the camera-ready version of the paper, from using author names when referring to citations, to making statements throughout the paper more clear (following the points raised in part 8 of your review).

**R3 –** Refer to the plot above for comparison with Network Slimming, which we will add to the paper. We believe that characterizing what aspects of a pruning method yield better winning tickets is an exciting and valuable research question. We hope that our work further motivates research on this question by introducing a second methodology via which to perform ticket search - one that is fundamentally distinct from IMP and relies on different tools.

[1] Christos Louizos, Max Welling, Diederik P. Kingma. Learning Sparse Neural Networks through $L_0$ Regularization. ICLR, 2018.
[2] Alex Renda, Jonathan Frankle, Michael Carbin. Comparing Rewinding and Fine-tuning in Neural Network Pruning. ICLR, 2020.
[3] Suraj Srinivas, Akshayvarun Subramanya, R. Venkatesh Babu. Training Sparse Neural Networks. CVPR Workshops, 2017.
[4] Trevor Gale, Erich Elsen, Sara Hooker. The State of Sparsity in Deep Neural Networks. ICML Workshops, 2019.


[Meta-Review · NeurIPS 2020]

The paper focuses on the recently popular Lottery Ticket Hypotheses (LTH). It provides a novel algorithm, Continuous Sparsification (CS), that finds the sparse "matching subnetworks" that can be trained in isolation to the full accuracy starting from early stage in training. Currently, the only existing method for performing this task is the so-called Iterative Magnitude Pruning (IPM). CS finds matching subnetworks that are sparser than results of IMP, and CS runs much faster (provided certain parallel resources). The method is based on a clear and easy-to-understand idea of introducing one new parameter per each weight of the network that is used to eliminate the weights. This is a significant and important contribution to the field of LTH. As pointed out by Rev#2, "it makes it possible for us to efficiently scale up the lottery ticket observations". Quoting the same reviewer, "The paper has a clear goal and conducts thorough experiments to demonstrate that the proposed technique meets that goal". Finally, the paper "follows all of the empirical best practices". In light of this, I recommend the acceptance of the paper. However, as agreed in the rebuttal (and based on later post-rebuttal discussions with the reviewers), in the final revision the authors should fully address the following list of action items. Below I include the summary of these action items; I will ask the authors to refer to the detailed updated reviews and address exact suggestions described there. *** 1. Writing quality. *** Improve writing quality in Sections 1 and 2. The authors have promised "a subsection introducing the reader to the nomenclature and precise definitions." *** 2. On the relationship between CS and L0 regularization method. *** Considering that all four reviewers were confused on this point, the authors need to update the camera ready version to feature the discussion appearing in the rebuttal in place of the brief, hand-wavey comparisons that are currently in the paper. *** 3. Comparison to standard pruning methods. This is IMPORTANT. *** Here, by pruning we mean methods that start from the trained dense networks and prune them. If "ticket search is a strictly more general and harder task than pruning," then there is no excuse for the paper not to feature comparisons to standard pruning methods. As such, the authors should prominently feature comparisons to these pruning methods, including already existing comparisons to STR and DNW, as well as additional comparison to the standard magnitude pruning (Gupta,2017). See post-rebuttal notes of Rev#1 and Rev#2 for exact suggestions. I would like to highlight once again, that this point is important. The authors seem to slightly disagree in the rebuttal, however all the reviewers are convinced that this point should be addressed properly. *** 4. Comparison to rigged lottery, 5x training. *** The authors compare their method to the Rigged Lottery (RigL) method. RigL does end-to-end sparse training, i.e. it starts with a random sparse network and *never* needs dense resources. Therefore, comparing resource-demanding CS with RigL is not exactly fair. The authors should either remove the current misleading comparison to RigL, or include the 5x (5 times longer) training results for RigL, if they like to compare fairly. Since the training in RigL is sparse, total training FLOPs needed for a 5x training is still less than 1x of pruning training usually. * * * * * * * * * * * * All the reviewers think (and I agree) that these items can be comfortably addressed before the camera ready deadline. I also believe that addressing them will make the submission significantly stronger and reader-friendly. In summary, I recommend the acceptance and trust the authors to address the (minor but extremely important) action items above.